# STEER: Simple Temporal Regularization For Neural ODEs

**Arnab Ghosh**
University of Oxford
arnabg@robots.ox.ac.uk

**Harkirat Singh Behl**
University of Oxford
harkirat@robots.ox.ac.uk

**Emilien Dupont**
University of Oxford
emilien.dupont@stats.ox.ac.uk

**Philip H. S. Torr**
University of Oxford
phst@robots.ox.ac.uk

**Vinay Namboodiri**
University of Bath
vpn22@bath.ac.uk

## Abstract

Training Neural Ordinary Differential Equations (ODEs) is often computationally expensive. Indeed, computing the forward pass of such models involves solving an ODE which can become arbitrarily complex during training. Recent works have shown that regularizing the dynamics of the ODE can partially alleviate this. In this paper we propose a new regularization technique: randomly sampling the end time of the ODE during training. The proposed regularization is simple to implement, has negligible overhead and is effective across a wide variety of tasks. Further, the technique is orthogonal to several other methods proposed to regularize the dynamics of ODEs and as such can be used in conjunction with them. We show through experiments on normalizing flows, time series models and image recognition that the proposed regularization can significantly decrease training time and even improve performance over baseline models.

## 1 Introduction

Neural Ordinary Differential Equations (ODE) are an elegant approach for building continuous depth neural networks [4]. They offer various advantages over traditional neural networks, for example a constant memory cost as a function of depth and invertibility by construction. As they are inspired by dynamical systems, they can also take advantage of the vast amount of mathematical techniques developed in this field. Incorporating continuous depth also enables efficient density estimation [18] and modeling of irregularly sampled time series [46].

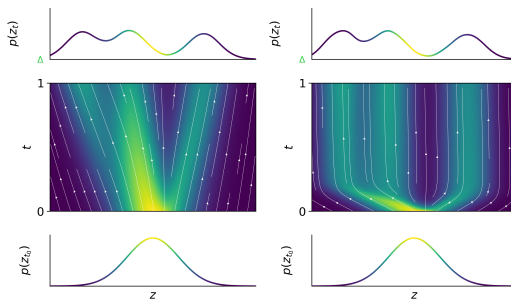

Figure 1: Flow fields of a standard Neural ODE (left) vs one equipped with STEER (right). These models transform a simple initial distribution $p(z_{t_0})$ to the target distribution $p(z_t)$. STEER learns simpler flows, which enable faster training.

The aforementioned advantages come at the expense of a significantly increased computational cost compared with traditional models like Residual Networks [21]. The training dynamics of Neural ODEs become more complicated as training progresses and the number of steps taken by the adaptive ODE solver (typically referred to as the number of function evaluations) can become prohibitively large [14]. In fact, several works have argued that this is one of the most

pressing limitations of Neural ODEs [4, 14, 16]. It is therefore important to design methods that can help reduce the computational cost of this tool.

A natural approach for tackling this issue is to explore regularization techniques that simplify the underlying dynamics for these models. Indeed, simpler dynamics have shown benefits in terms of faster convergence and a lower number of function evaluations both during training and testing [14]. [16] recently proposed an interesting technique for explicitly regularizing the dynamics based on ideas from optimal transport showing significant improvements in the effective convergence time. In this work, we propose a regularization technique in order to further accelerate training. Our regularization is orthogonal to previous approaches and can therefore be used in conjunction with them. Our technique is extremely simple: during training we randomly sample the upper limit of the ODE integration.

Specifically, the solver in Neural ODEs takes as input the integration limits $(t_0, t_1)$. We propose to introduce regularization by randomly perturbing the end time parameter $t_1$ to be $t_1 \pm \delta$ where $\delta$ is a random variable. Empirically, we demonstrate that, in a wide variety of tasks, the proposed regularization achieves comparable performance to unregularized models at a significantly lower computational cost. We test our regularization technique on continuous normalizing flows [18], irregularly sampled time series [46] and image recognition tasks and show that it improves performance across all these tasks. It also performs better in certain instances of Stiff ODEs. Stiff ODEs are a widely studied topic in the field of engineering[6]. During training the dynamics learnt by the implicit model parametrized by the neural network can turn into an instance of a Stiff ODE. Techniques which could tackle the instances of Stiff ODE would prove to be instrumental as these implicit deep learning models mature.

## 2   Related Work

Connections between deep neural networks and dynamical systems were made as early as [31]. The adjoint sensitivity technique was explored in [44]. Some early libraries such as dolfin [15, 45], implemented the adjoint method for neural network frameworks. Previous works have shown the connection between residual networks and dynamical systems [47, 19, 36, 53, 51, 1]. By randomly dropping some blocks of a residual network, it was demonstrated that deep neural networks could be trained [23] or tested [50, 17] with stochastic depth, and that such a training procedure produces ensembles of shallower networks.

Neural ODEs [4] introduced a framework for training ODE models with backpropagation. Recent works [38, 58, 5, 25, 37, 9, 20, 40] have tried to analyze this framework theoretically and empirically. Neural ODEs were extended to work with flow based generative models [18]. In contrast to standard flow based models such as [29, 11, 43, 42], continuous normalizing flows do not put any restrictions on the model architecture. Furthermore, Neural ODEs have been applied in the context of irregularly sampled time series models [46], reinforcement learning [13] and for set modeling[34]. Neural ODEs have also been explored for harder tasks such as video generation [57, 27] and for modeling the evolving dynamics of graph neural networks [55]. Recent work has also extended Neural ODEs to stochastic differential equations [33, 35, 22, 41, 54, 26].

The methods most relevant to our work are [28, 16] and [14]. A regularization based on principles of optimal transport which simplifies the dynamics of continuous normalizing flow models, was introduced in [16] to make training faster. Equations which are easier to learn were analyzed in [28] whereby they used a differential surrogate for the time cost of standard numerical solvers, using higher-order derivatives of solution trajectories. Augmented Neural ODEs [14] introduced a simple modification to Neural ODEs by adding auxillary dimensions to the ODE function and showing how this simplified the dynamics. Our work is complementary to these methods, and we empirically demonstrate that our method can be used to further improve the performance of these techniques.

There is a huge treasure of work which deals with instability in training ODEs[8, 7]. Curtiss et al. [6] were one of the first works that dealt with solving stiff equations. It was observed in Curtiss and Hirschfelder [6] that explicit methods failed for the numerical solution of ordinary differential equations that model certain chemical reactions. It is very difficult to integrate "stiff" equations by ordinary numerical methods. Small errors are rapidly magnified if the equations are integrated. The other case might be that the errors oscillate rapidly. Implicit methods such as the second order Adams-Moulton (AM2) [52] are elegant techniques which deal with some of the problems of stiff ODEs. In the context of Neural ODEs[4] implicit computations reduces the efficiency of the solvers.

This effect is especially apparent in complex modeling tasks such as generative modeling using continuous normalizing flows[18] and irregularly sampled timeseries models[46]. Hence an analysis of simple scenarios of stiff equations which cause problems for standard Neural ODEs are discussed in the following sections.

## 3 Methodology

### 3.1 Neural ODEs

Neural ODEs can be interpreted as a continuous equivalent of Residual networks [21]. The transformation of a residual layer is $\mathbf{z}_{t+1} = \mathbf{z}_t + f_t(\mathbf{z}_t)$, where the representation at layer $t$ is $\mathbf{z}_t$ and $f_t(\mathbf{z}_t)$ is a function parametrized by a neural network. The output dimension of the function $f_t(\mathbf{z}_t)$ should be the same as the input dimension to allow it to be added to the previous layer's features $\mathbf{z}_t$. This transformation can be rewritten as

$$\frac{\mathbf{z}_{t+\delta_t} - \mathbf{z}_t}{\delta_t} = f_t(\mathbf{z}_t), \quad \delta_t = 1, \tag{1}$$

which can be interpreted as finite difference approximation of the derivative. In contrast to traditional deep neural networks where $\delta_t = 1$ is fixed, Neural ODEs [4] introduced a continuous version by analysing Equation 1 in the limit $\delta_t \to 0$. The resulting equation becomes

$$\frac{d\mathbf{z}(t)}{dt} = f(\mathbf{z}(t), t). \tag{2}$$

To transform an input $\mathbf{x}$ to an output $\mathbf{y}$, Residual networks use a sequence of $k$ functions to transform the representation. In an analogous manner, Neural ODEs evolve the representation from the initial time $t = t_0$, $\mathbf{z}(t_0) = \mathbf{x}$, to the final time $t = t_1$, $\mathbf{z}(t_1) = \mathbf{y}$. The representation $\mathbf{z}(t)$ at any given time $t$ can then be obtained by solving Equation 2 using an off-the-shelf ODE solver with the function $f(\mathbf{z}(t), t)$ parameterized by the neural network. To update the parameters of the function $f(\mathbf{z}(t), t)$ such that $\mathbf{z}(t_1) \approx \mathbf{y}$ we backpropagate through the operations of the ODE solver to obtain the appropriate gradients.

### 3.2 Temporal Regularization

The formulation of a standard Neural ODE is given by

$$\mathbf{z}(t_1) = \mathbf{z}(t_0) + \int_{t_0}^{t_1} f(\mathbf{z}(t), t, \theta)dt = \text{ODESolve}(\mathbf{z}(t_0), f, t_0, t_1, \theta), \tag{3}$$

where $\mathbf{z}(t)$ is the latent representation at any time $t$. $\theta$ are the parameters of the function $f$ and $(t_0, t_1)$ are the limits of the integration. The initial condition of the ODE is set as $\mathbf{z}(t_0) = \mathbf{x}$.

Neural ODEs often take an excessively large time to train, especially in the case of continuous normalizing flows [18] . The dynamics become more complicated as training progresses thus leading to an increase in the number of function evaluations. In the context of Neural ODEs, the dynamics of the ODE and training time are intricately related. In practice, simpler dynamics [14, 16] often lead to faster training with fewer function evaluations. Regularization is commonly used in machine learning to ease the training of a model and improve its generalization. Some common instances include Batchnorm [24] and early stopping [56].

Neural ODEs continuously evolve the representation $\mathbf{z}(t)$ with time $t$. Thus in the context of neural ODEs, an alternative type of regularization is possible, namely temporal regularization. In this work, we propose a simple regularization technique that exploits this temporal property. We propose to perturb the final time $t_1$ for the integration of the Neural ODE in Equation 3 during training. At each training iteration, the final time $t_1$ for the integration is stochastically sampled. More formally, if we

assume $t_0 < t_1$, then our regularization can be formulated as

$$\mathbf{z}(t_1) = \mathbf{z}(t_0) + \int_{t_0}^{T} f(\mathbf{z}(t), t, \theta) dt = \text{ODESolve}(\mathbf{z}(t_0), f, t_0, T, \theta),$$
$$T \sim \text{Uniform}(t_1 - b, t_1 + b),$$
$$b < t_1 - t_0 : \text{parameter controlling interval of uniform distribution.}$$
(4)

We call the technique STEER, which stands for Simple Temporal Regularization. Since Neural ODEs model a continuous series of transformations, applying STEER is essentially equivalent to performing a variable number of transformations on the input $\mathbf{x}$ in each training iteration, thereby introducing a new form of stochasticity in the training process.

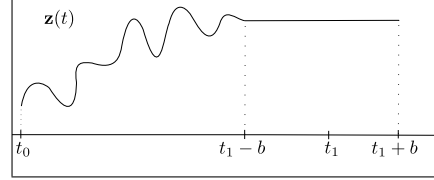

We observe that the proposed regularization effectively ensures convergence to the solution at time $t_1 - b$. The effective behavior of $\mathbf{z}(t)$ can be see in Fig. 2

Figure 2: Effective behavior of Neural ODEs with STEER regularization. The solution is reached at a time $t_1 - b$ instead of time $t_1$.

### 3.3 Stiff ODEs

Stiff ODEs have been extensively researched in the field of differential equations [2]. Inspite of years of research a concrete definition of Stiff ODEs has not been agreed upon [49]. We highlight how some of the problems of Stiff ODEs could impact the training dynamics of Neural ODEs in specific scenarios. Certain discussions of Stiff ODEs define that the stiffness concerns the underlying mathematical form of the equation, integration time and a set of initial data.

In terms of the mathematical equation, stiffness is a problem that can arise when numerically solving ODEs. In such systems there is typically a fast and a slow transient component which causes adaptive ODE solvers to take exceedingly small steps. This increases the number of function evaluations and the time taken to integrate the function. Furthermore, in Neural ODEs, the function $\frac{d\mathbf{z}}{dt}$ is parameterized by a neural network which is updated during training, and could become stiff. This would be hard for a traditional ODE solver to integrate efficiently, thereby making this a limitation.

Stiffness ratio is one of the techniques used to characterize some forms of stiff equations. The stiffness ratio is used to characterize the stiffness for the general case of linear constant coefficient systems. These systems can be represented as

$$\frac{d\mathbf{z}}{dt} = \mathbf{A}\mathbf{z} + f(t),$$
(5)

where $\mathbf{z}, f(t) \in \mathbb{R}^n$ and $\mathbf{A}$ is a $n \times n$ matrix with eigenvalues $\lambda_i \in \mathbb{C}, i = 1, 2, \ldots, n$ (assumed distinct) and corresponding eigenvectors $\mathbf{c_i} \in \mathbb{C}^n, i = 1, 2, \ldots, n$. The eigenvalues and eigenvectors can be complex. The general solution of Equation 9 takes the form

$$\mathbf{z}(t) = \sum_{i=1}^{n} \kappa_i \exp(\lambda_i t) \mathbf{c}_i + \mathbf{g}(t).$$
(6)

If $\lambda_i$ is complex, the term $\exp(\lambda_i t)$ varies sinusoidally. The stiffness ratio is defined as $\frac{\max_{i=1\ldots n}|Re(\lambda_i)|}{\min_{i=1\ldots n}|Re(\lambda_i)|}$, where $Re(\lambda_i)$ denotes the real component of $\lambda_i \in \mathbb{C}$. Ordinarily, temporal regularization might fail in the face of stiffness. However, in our empirical experiments we observed that at least in the case of toy problems, STEER actually performed better. This analysis is provided in the next section.

## 4 Experiments

In this section, we demonstrate the effectiveness of the proposed method on several classes of problems. First, we conduct experiments on continuous normalizing flow based generative models,

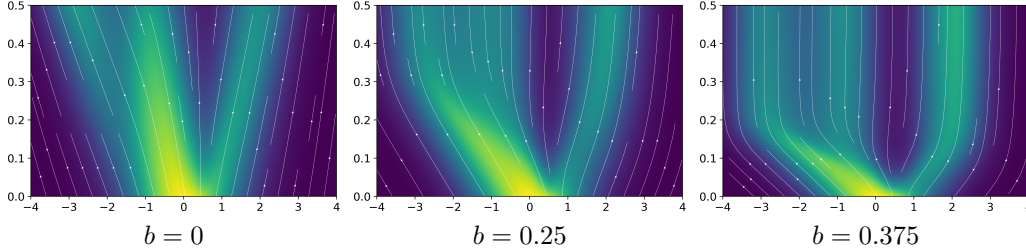

$$b = 0 \qquad\qquad b = 0.25 \qquad\qquad b = 0.375$$

Figure 3: Plots showing the path taken by FFJORD with STEER regularization. STEER regularization ($b = 0.25$ and $b = 0.375$) performs the same transformation in shorter paths.

and compare against FFJORD [18] and RNODE [16]. Then, we conduct experiments on irregularly sampled time-series models, and compare against the models and baselines proposed in Latent ODEs [46]. We also conduct experiments on feedforward models for image recognition, and compare against Neural ODEs and Augmented Neural ODEs [14]. We finally evaluate STEER and standard Neural ODEs on a toy stiff ODE experiment.

## 4.1 Generative Models

Continuous normalizing flow based generative models [18, 4] provide an alternate perspective to generative modeling. Let $p(\mathbf{x})$ be the likelihood of a datapoint $\mathbf{x}$ and $p(\mathbf{z})$ be the likelihood of the latent $\mathbf{z}$. FFJORD [18] showed that under certain conditions the change in log probability also follows a differential equation $\frac{d \log(p(\mathbf{z}(t)))}{dt} = -\mathrm{tr}\left(\frac{df}{d\mathbf{z}(t)}\right)$. Thus the total change in log probability can be obtained by integrating the equation which leads to

$$\log p(\mathbf{z}(t_1)) = \log p(\mathbf{z}(t_0)) - \int_{t_0}^{t_1} \mathrm{tr}\left(\frac{df}{d\mathbf{z}(t)}\right) dt. \tag{7}$$

This allows to compute the point $\mathbf{z_0}$ which generates a given datapoint $\mathbf{x}$, and $\log p(\mathbf{x})$ under the model by solving the initial value problem:

$$\underbrace{\begin{bmatrix} \mathbf{z_0} \\ \log p(\mathbf{x}) - \log p_{\mathbf{z_0}}(\mathbf{z_0}) \end{bmatrix}}_{\text{solutions}} = \int_T^{t_0} \underbrace{\begin{bmatrix} f(\mathbf{z}(t), t; \theta) \\ -\mathrm{tr}\left(\frac{df}{d\mathbf{z}(t)}\right) \end{bmatrix}}_{\text{dynamics}} dt, \qquad \underbrace{\begin{bmatrix} \mathbf{z}(T) \\ \log p(\mathbf{x}) - \log p(\mathbf{z}(T)) \end{bmatrix}}_{\text{initial values}} = \begin{bmatrix} \mathbf{x} \\ \mathbf{0} \end{bmatrix}. \tag{8}$$

The combined dynamics of $\mathbf{z}(t)$ and the log-density of the sample is integrated backwards in time from $T$ to $t_0$. The log likelihood $\log p(\mathbf{x})$ can be computed by adding $\log p_{\mathbf{z_0}}(\mathbf{z_0})$ to the solution of Equation 8. A major drawback of continuous normalizing flow based models is extremely slow convergence. STEER regularization enables faster convergence by sampling $T \sim \mathrm{Uniform}(t_1 - b, t_1 + b)$ at each iteration.

We focus on the task of density estimation of image datasets using the same architectures and experimental setup as FFJORD [18]. More details are provided in the supplementary material. The performance of the model is evaluated on multiple datasets, namely MNIST [32], CIFAR-10 [30] and ImageNet-32 [10], and the results are shown in Table 1. The performance is also analyzed in the simple scenario of a 1D Mixture of Gaussians as shown in Fig. 3.

It can be seen in Table 1 that models with STEER regularization converge much faster. In some cases, such as on CIFAR-10, our model is $30\%$ faster than the state of the art.

Fig. 3 shows the evolution of the probability distribution from the latent distribution $p(\mathbf{z})$ to the data distribution $p(\mathbf{x})$ at different test times. It can be seen that the trajectories reach the final states at an earlier test time with STEER regularization.

## 4.2 Times Series Models

For time series models, the function is evaluated at multiple times $t_0, ..t_n$. We employ the regularization between every consecutive pair of points $(t_i, t_{i+1})$. More formally, if the time is split into

|  | MNIST | | CIFAR-10 | | IMAGENET-32 | |
|---|---|---|---|---|---|---|
|  | BITS/DIM | TIME | BITS/DIM | TIME | BITS/DIM | TIME |
| FFJORD, VANILLA. | 0.974 | 68.47 | 3.40 | $> 5$ days | 3.96 | $> 5$ days |
| FFJORD RNODE. [16] | 0.973 | 24.37 | 3.398 | 31.84 | 2.36 | 30.1 |
| FFJORD, VANILLA. + STEER ($b = 0.5$) | 0.974 | 51.21 | 3.40 | 86.34 | 3.84 | $> 5$ days |
| FFJORD, VANILLA. + STEER ($b = 0.25$) | 0.976 | 58.21 | 3.41 | 103.4 | 3.87 | $> 5$ days |
| FFJORD RNODE + STEER ($b = 0.5$) | **0.971** | **16.32** | **3.397** | **22.24** | **2.35** | **24.9** |
| FFJORD RNODE + STEER ($b = 0.25$) | 0.972 | 19.71 | 3.401 | 25.24 | 2.37 | 27.84 |

Table 1: Test accuracy and training time comparison on various image datasets. Unless mentioned as days, the time reported is in hours.

| Model | Accuracy |
|---|---|
| Latent ODE (ODE enc) | $0.846 \pm 0.013$ |
| Latent ODE (RNN enc.) | $0.835 \pm 0.010$ |
| ODE-RNN | $0.829 \pm 0.016$ |
| Latent ODE (ODE enc, STEER) | **$0.880 \pm 0.011$** |
| Latent ODE (RNN enc, STEER) | $0.865 \pm 0.021$ |
| ODE-RNN (STEER) | $0.872 \pm 0.012$ |

Table 2: Per-time-point classification accuracies on the Human Activity dataset.

| Model | AUC |
|---|---|
| Latent ODE (ODE enc) | $0.829 \pm 0.004$ |
| Latent ODE (RNN enc.) | $0.781 \pm 0.018$ |
| ODE-RNN | **$0.833 \pm 0.009$** |
| Latent ODE (ODE enc, STEER) | $0.810 \pm 0.018$ |
| Latent ODE (RNN enc, STEER) | $0.772 \pm 0.014$ |
| ODE-RNN (STEER) | $0.811 \pm 0.007$ |

Table 3: Per-sequence classification AUC on Physionet.

consecutive intervals $(t_0, t_1)...(t_{n-1}, t_n)$, we start with $t_0$ and the initialization $\mathbf{z}(t_0)$ and sample the end time as $T \sim \text{Uniform}(t_1 - b, t_1 + b)$. The resulting integration using the ODE solver gives us the latent representation $\mathbf{z}(t_1)$. We repeat the same procedure where the limits of integration are now $(t_1, t_2)$. In this iteration $t_1$ is not altered while $T \sim \text{Uniform}(t_2 - b, t_2 + b)$ is sampled to obtain $\mathbf{z}(t_2)$ and so on. In the case of irregularly sampled time series models, the length of the interval $t_{i+1} - t_i$ can itself vary. The parameter $b$ in this case is adaptive such that $b = |t_{i+1} - t_i| - \epsilon$. The $\epsilon$ parameter is added to avoid numerical instability issues. In the case of time series models the parameter $b$ changes at each forward pass thus adding another source of stochasticity.

Latent ODEs [46] introduced irregularly sampled time series models in which Neural ODEs perform much better. Indeed these datasets break the basic assumptions of a standard RNN, namely that the data are not necessarily collected at regular time intervals. This is the case in several real life scenarios such as patient records or human activity recognition which are observed at irregularly sampled timepoints.

The Mujoco experiments consist of a physical simulation of the "Hopper" model from the Deepmind control suite. The initial condition uniquely determines the trajectory. It consists of $10,000$ sequences of $100$ time points each. The Human Activity dataset [46] consists of sensor data fitted to individuals. The sensor data is collected using tags attached to the belt, chest and ankles of human performers. It results in $12$ features in total. The task involves classifying each point as to which activity was being performed. Physionet [48] consists of time series data from initial $48$ hours of ICU admissions. It consists of sparse measurements where only some of the features' measurements are taken at an instant. The measurement frequency is also irregular which makes the dataset quite challenging. To

|  | Interpolation (% Observed Pts.) | | | | Extrapolation (% Observed Pts.) | | | |
|---|---|---|---|---|---|---|---|---|
| Model | 10 % | 20 % | 30 % | 50 % | 10 % | 20 % | 30 % | 50 % |
| Latent ODE (ODE enc) | 0.360 | 0.295 | 0.300 | 0.285 | 1.441 | 1.400 | 1.175 | 1.258 |
| Latent ODE (RNN enc.) | 2.477 | 0.578 | 2.768 | 0.447 | 1.663 | 1.653 | 1.485 | 1.377 |
| ODE-RNN | 1.647 | 1.209 | 0.986 | 0.665 | 13.508 | 31.950 | 15.465 | 26.463 |
| Latent ODE (ODE enc, STEER) | **0.230** | **0.235** | **0.246** | **0.210** | **1.01** | **0.98** | **1.01** | **1.04** |
| Latent ODE (RNN enc., STEER) | 2.31 | 0.561 | 2.534 | 0.386 | 1.541 | 1.498 | 1.236 | 1.119 |
| ODE-RNN (STEER) | 1.528 | 1.09 | 0.796 | 0.603 | 11.508 | 29.34 | 14.23 | 23.528 |

Table 4: Test Mean Squared Error (MSE) ($\times 10^{-2}$) on the MuJoCo dataset.

|          | NODE              | NODE(STEER)       | ANODE             | ANODE(STEER)        |
|----------|-------------------|-------------------|-------------------|---------------------|
| MNIST    | $97.2\% \pm 0.2$  | $98.3\% \pm 0.3$  | $98.2\% \pm 0.1$  | $\mathbf{98.6\% \pm 0.3}$ |
| CIFAR-10 | $53.7\% \pm 0.4$  | $54.9\% \pm 0.5$  | $60.6\% \pm 0.4$  | $\mathbf{62.1\% \pm 0.2}$ |
| SVHN     | $81.0\% \pm 0.6$  | $81.8\% \pm 0.3$  | $83.5\% \pm 0.5$  | $\mathbf{84.1 \pm 0.2}$ |

Table 5: Test accuracies on Feedforward models on various datsets.

predict mortality, binary classifiers were constructed by passing the hidden state to a 2 layer binary classifier.

We use the same architecture and experiment settings as used in Latent ODEs [46]. For the Mujoco experiment 15 and 30 latent dimensions were used for the generative and recognition models respectively. The ODE function had 500 units in each of the 3 layers. A 15 dimensional latent state was used for the Human Activity dataset. For the Physionet experiments the ODE function has 3 layers with 50 units each. The classifier consisted of 2 layers with 300 units each.

In the case of irregularly sampled timeseries with Latent ODEs, the number of function evaluations is not a major concern since the training time is quite small, so we compare only in terms of accuracy. It can be seen in Table 2 and Table 4 that STEER regularization consistently improves the performance of ODE based models. Table 4 demonstrates that the extrapolation results are much better with STEER regularization, indicating that it may help improve the generalization of the model. However in the case of Physionet Table 3, the results do not improve over standard Neural ODEs, although the performance remains very close.

## 4.3 Feedforward Models

In feedforward learning for classification and regression, the Neural ODE framework can be used to map the input data $\mathbf{x} = \mathbf{z}(t_0) \in \mathbb{R}^d$ to features $\mathbf{z}(t) \in \mathbb{R}^d$ at any given time $t$. For standard neural ODEs, one integrates in the range $(t_0, t_1)$, and for STEER the integration is performed in the range $(t_0, T)$ where $T \in \text{Uniform}(t_1 - b, t_1 + b)$. Once the features $\mathbf{z}(T)$ at the final time $T$ are obtained, a linear layer $\mathbb{R}^d \to \mathbb{R}$ is used to map the features to the correct dimension for downstream tasks such as classification or regression.

We evaluate STEER on a similar set of experiments as performed in [14] on the MNIST [32], CIFAR-10 [30] and SVHN [39] datasets using convolutional architectures for $f(\mathbf{z}(t), t)$. We use the same architectures and experimental settings as used in [14] for both Neural ODEs (NODE) and Augmented Neural ODEs (ANODE). For the MNIST experiments, 92 filters were used for NODEs while 64 filters were used for ANODEs to compensate for 5 augmented dimensions. For the CIFAR-10 and SVHN experiments, 125 filters were used for NODEs while 64 filters were used for ANODEs with 10 augmented dimensions. Such a design was employed to ensure a roughly similar number of parameters for a fair comparison.

Table 5 demonstrates the performance of STEER regularization with $b = 0.99$. STEER provides a small boost in accuracy and performs consistently across all datasets.

## 4.4 A Toy Example of Stiff ODE

In this experiment, we analyze a toy example of stiff ODE from [3]. The aim of this experiment is to highlight the vulnerability of Neural ODEs to stiffness. The experimental setting for section 4.4 was similar to the one used in the official github repository [1] of Neural ODE. Instead of using a 2-D ODE we used a 1-D ODE to be able to better visualize and analyze the behavior. We used Eq 12 in the paper as mentioned in the numerical methods book by [3] on page 752. It was surprising that a Neural ODE was able to model a 2 dimensional model with ease while the stiff equation was extremely hard for the Neural ODE to model.

The ODE is given by

$$\frac{dy}{dt} = -1000y + 3000 - 2000e^{-t}, \tag{9}$$

with the initial condition of $y(0) = 0$. The solution for the above stiff ODE is $y = 3 - 0.998e^{-1000t} - 2.002e^{-t}$. We use a simple architecture with a single hidden layer of dimension $500$ for both the standard Neural ODE and the one with STEER. The effective input dimension is 2 resulting by concatenating $y$ and $t$, and the output dimension is 1.

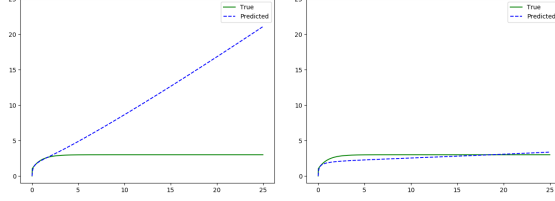

Figure 4: Standard Neural ODE struggles while STEER regularization helps to fit a stiff ODE with stiffness ratio $r = 1000$.

As shown in Fig. 4, the solution is initially dominated by the fast exponential term $e^{-1000t}$. After a short period ($t < 0.005$), this transient dies down and the solution is dictated by the slow exponential $e^{-t}$. The training interval $[0, 15]$ is broken down into small intervals of length $(t_1 - t_0) = 0.125$. We sample 1000 random initial points $t_0$ and use $y(t_0)$ as the initialization and then try to estimate the value of the function $y(t_1)$ at $t_1 = t_0 + 0.125$. It can be seen in Fig. 4 that standard Neural ODE doesn't show the same asymptotic behavior as the actual solution, whereas STEER regularization enables better asymptotic behavior. The minimum Mean Squared Error(MSE) loss over all the points of the trajectory in the test interval $[0, 25]$ for standard Neural ODE is 3.87, and with STEER regularization with $b = 0.124$ it is 0.52.

To further study the behaviour with varying stiffness ratio, the formula in Equation 9 is generalized as $\frac{dy}{dt} = -ry + 3r - 2re^{-t}$. The generalized equation has the same asymptotic behavior as the original. It also reaches a steady state at $y = 3$. The $-ry$ term would introduce a term of $e^{-rt}$ and another term of $e^{-t}$ would be present in the solution. Thus the stiffness ratio as described in Subsection 3.3 is exactly $r$. The plot in Fig. 5 shows that as the stiffness increases, the STEER regularization compares better in terms of MSE to models without the regularization. The experiments were performed using the dormand-prince [12] ODE solver.

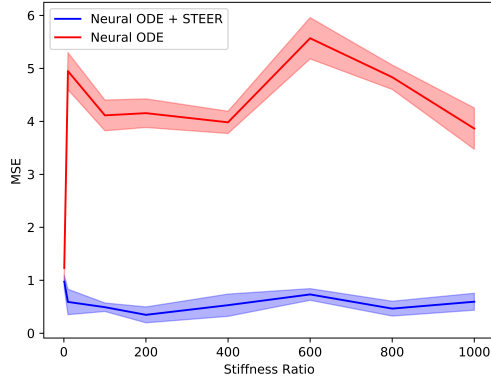

Figure 5: Model with STEER regularization achieves a much lower MSE on a Stiff ODE as the stiffness ratio increases.

While we do not promise to have proposed a solution for the difficult problem of Stiff-ODEs [3], we empirically observed that it gives improvement on toy examples. We want to highlight that further research is needed in this important area of Stiffness. We show analysis about the choice of $b$ for our technique in the appendix.

## 4.5 A Toy example of a Timeseries Model:

In this experiment, we analyze a toy example of irregularly sampled timeseries models. The experimental setting was the same as in [46]. The models were tested on a toy dataset of periodic trajectories. The initial point was sampled from a standard Gaussian. Gaussian noise was added to the subsequent observations. Each trajectory has a 100 observations. A fixed subset of these points from each trajectory are sampled. The entire set of 100 observations are attempted to be reconstructed. In the original experiment by [46] the amplitude was the same throughout the trajectories while the frequency was varied for the $A.sin(\phi t)$ where $\phi$ is the frequency of the sine component. We made a simple change to the experimental setting whereby we also added a $t$ component to the sampled periodic trajectories. Thus the final form of the trajectories being sampled are $A.sin(\phi t) + t$ where $A$ is constant and $\phi$ can vary within the sampled trajectories. In this experimental setting we see that standard Neural ODEs start to produce trajectories which are qualitatively quite different from the actual sampled trajectories. STEER on the other hand produces trajectories which match the qualitative behavior of the sampled trajectories more closely.

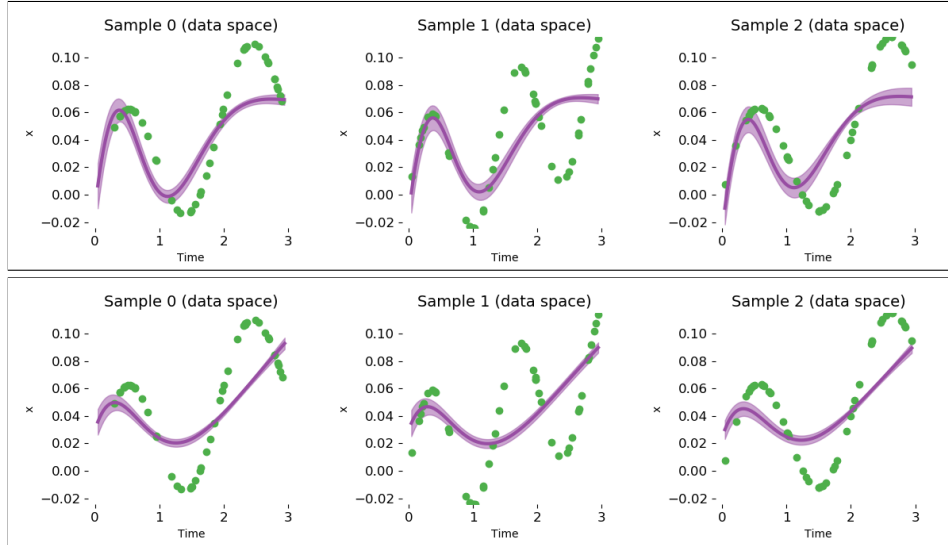

Figure 6: A simple addition of $t$ to $A.sin(\phi t)$ changes the qualitative performance of Neural ODE vs STEER. The top row demonstrates the 3 sample trajectories that have been irregularly sampled using STEER, while the bottom row is with a standard Neural ODE. The green points indicate the irregularly sampled points, while the rest of the points are predicted using each trained model. The amplitude $A$ of the model is the same across the various samples but the frequency $\phi$ is variable across the samples. As we see from the sampled trajectories STEER models the qualitative behavior of the trajectories better than standard Neural ODEs.

# 5    Conclusion

In this paper we introduced a simple temporal regularization method for Neural ODEs. The regularization consists of perturbing the final time of the integration of the ODE solver. The main advantage of our work is the empirical evidence showing the regularization results in simpler dynamics for Neural ODEs models on a wide variety of tasks. This is demonstrated for a specific stiff ODE that is challenging for a basic Neural ODE solver, faster continuous normalizing flow models and improved or comparable accuracy for time series models and deep feedforward networks. We further show that our approach is orthogonal to other regularization approaches and can be thus combined with them to further improve Neural ODEs.

# 6    Broader Impact

Neural ODEs operate under the broad umbrella of 'Implicit Methods in Deep Learning'. The last decade in deep learning was characterized by layer based deep neural networks. The limits of that framework are starting to approach whereby one can fit only so many layers in the GPU of a computer. Thus, alternate forms of computational frameworks are emerging whereby the function is not explicitly modeled. Rather a pseudo function is learnt from which the original function that needed to be approximated can be retrieved. This shows a different model of computation and previous work has shown signs that it is much more parameter efficient than resnet based deep neural networks.

Our work is in the broad area of Neural ODEs which provides continuous depth neural networks. This line of work would have a similar impact as most supervised deep learning techniques. However,current continuous depth neural networks show limitations that are improved through our proposed regularization scheme. Thus, our models converge faster during training by reducing the number of function evaluations and effective training time. The resulting trained models also require fewer function evaluations at inference, thus reducing the computation at test time too. Our research could potentially encourage the search for more such efficient techniques for training neural ODEs.

Although we have highlighted some drawbacks of existing Neural ODEs in the context of Stiff ODEs. We do not fully understand why our solution works better in the case of stiff ODEs. The issue of stiff ODEs has a rich history of research in engineering. Inevitably during training, there is a possibility that the dynamics learnt by the neural network might be an instance of a stiff equation. This cannot be prevented with the current formulation. As Neural ODEs become more stable and mature we need to understand the effects of stiffness. These models can be truly useful if we understand it from all possible points of failure.

## Acknowledgments and Disclosure of Funding

Arnab Ghosh was funded by the University of Oxford through a studentship using combined corporate gifts from Microsoft and Technicolor. Harkirat Singh Behl was supported using a Tencent studentship through the University of Oxford. Emilien Dupont was funded by the University of Oxford through a Deepmind studentship. Philip H.S. Torr was supported by the Royal Academy of Engineering under the Research Chair and Senior Research Fellowships scheme, EPSRC/MURI grant EP/N019474/1 and FiveAI. Vinay Namboodiri was funded using Startup funding support from University of Bath.

## Footnotes

[1]https://github.com/rtqichen/torchdiffeq/blob/master/examples/ode_demo.py

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
