[Supplementary Material]

# (Appendix) STEER: Simple Temporal Regularization For Neural ODEs

**Arnab Ghosh**
University of Oxford
arnabg@robots.ox.ac.uk

**Harkirat Singh Behl**
University of Oxford
harkirat@robots.ox.ac.uk

**Emilien Dupont**
University of Oxford
emilien.dupont@stats.ox.ac.uk

**Philip H. S. Torr**
University of Oxford
phst@robots.ox.ac.uk

**Vinay Namboodiri**
University of Bath
vpn22@bath.ac.uk

## 1  Modified Picard's Iteration:

Let the differential equation be

$$\frac{d\phi}{dt} = f(t, \phi(t)), \quad \phi(t_0) = z_0. \tag{1}$$

The modified Picard's iteration can be formulated as

$$\phi_0(t) = z_0, \tag{2a}$$

$$\phi_{k+1}(t) = z_0 + \int_{t_0}^{t+\delta_{k+1}} f(s, \phi_k(s))ds, \tag{2b}$$

$$\delta_{k+1} \sim \text{Uniform}(-b, b). \tag{2c}$$

Although the modified Picard's iteration is very close to our technique, it is not the exact same process that is simulated by our proposed technique. There are subtle differences which make the proposed technique different to the one simulated by the modified Picard's iteration. Picard's iteration constructs a sequence of approximate functions $\{\phi_k(t)\}$ which eventually converge to the desired solution. Equation 2a defines the initial approximation $\phi_0(t)$ as the initial condition $z_0$ of the ODE. Equation 2b describes the recurrence relation that relates $\phi_{k+1}(t)$ to $\phi_k(t)$. The recurrence relation adds a $\delta_{k+1}$ (Equation 2c) term which is randomly sampled.

It can be seen that the right-hand side Equation 2b defines an operator that maps a function $\phi$ to a function $T[\phi]$ as

$$T[\phi_k](t) = \phi_0 + \int_{t_0}^{t+\delta} f(s, \phi_k)ds, \quad \delta \sim \text{Uniform}(-b, b). \tag{3}$$

The following theorem shows the existence of a unique solution for the modified Picard's iteration for our method.

**Theorem 1.1.** *Suppose that $f : \mathbb{R}^2 \to \mathbb{R}$ satisfies the Lipschitz condition $|f(t, x_2) - f(t, x_1)| \leq L|x_2 - x_1|$ where $0 \leq L < \infty$. Suppose that $f$ is continuous and $\exists\ M$ such that $0 \leq M < \infty$, $|f(t, x)| \leq M$, $\forall(t, x)$. Then the sequence $\{\phi_k\}$ generated by the iteration $\phi_{k+1} = T[\phi_k], \phi_0(t) = z_0$ is a contraction in expectation.*

*The sequence $\{\phi_k(t)\}$ converges to a unique fixed point $\phi^*(t)$.*

Figure 1: Conditions of the proof.

*Proof.* Let the starting range of $t$ for the analysis be $(t_0 - \sigma, t_0 + \sigma)$. Let $c$ be such that $\forall i, |\phi_i(t) - \phi_0| \le c$. The solution only exists in the range $t \in (t_0 - \frac{a}{2}, t_0 + \frac{a}{2})$ where $a < \min(\frac{c}{M}, \frac{1}{2L})$. The parameter $b$ for STEER is chosen as $b \le \frac{a}{2}$ to ensure that the final effective time after sampling $\delta$ $t + \delta \in (t_0 - a, t_0 + a)$.

Let $\phi_{k+1}(t) = T[\phi_k] = \phi_0 + \int_{t_0}^{t+\delta} f(s, \phi_k)ds, \quad \delta \sim \text{Uniform}(-b, b)$ is well defined on $[t_0 - a, t_0 + a]$. $\phi_{k+1}(t)$ is continuous since both $\phi_k(t)$ and $f$ are continuous.

$\phi_{k+1}(t) \in \mathbb{R}$ since $|\phi_{k+1}(t) - \phi_0| = |\int_{t_0}^{t_1} f(s, \phi_k(s))ds| \le M|t - t_0| \le Ma < c$. This is by choice of $a$.

Let the metric on the space of solutions $\Phi$ be defined such that if $\Delta(\phi_k, \phi_{k+1}) = \max_{[t_0-a, t_0+a]} |\phi_k(t) - \phi_{k+1}(t)|$. $\Phi$ is a complete metric space which implies that all Cauchy sequences converge. We show using Lemma 1.2 that the operator $T$ is a contraction in expectation. Lemma 1.4 shows the convergence of the sequence of functions $\{\phi_k\}$. Finally Lemma 1.5 shows why the fixed point is unique with high probability. $\square$

**Lemma 1.2.** $\mathbb{E}_{|\delta_2| - |\delta_1|} \Delta(T\phi_1, T\phi_2) \le \frac{1}{2}\Delta(\phi_1, \phi_2)$

*Proof.*

$$|T\phi_2(t) - T\phi_1(t)|$$

$$= |\int_{t_0}^{t+\delta_2} (f(s,\phi_2(s))ds - \int_{t_0}^{t+\delta_1} (f(s,\phi_1(s)))ds|$$

$$= |\int_{t_0}^{t} (f(s,\phi_2(s) - f(s,\phi_1(s)))ds + \int_{t}^{t+\delta_2} (f(s,\phi_2(s)))ds - \int_{t}^{t+\delta_1} (f(s,\phi_1(s)))ds|$$

$$\leq |\int_{t_0}^{t} (f(s,\phi_2(s) - f(s,\phi_1(s)))ds + \int_{t}^{t+\delta_2} (f(s,\phi_2(s)))ds - \int_{t}^{t+\delta_1} (f(s,\phi_1(s)))ds|$$

$$\leq |L\int_{t_0}^{t} |\phi_2(s) - \phi_1(s)|ds + M|\delta_2| - M|\delta_1||$$

$$\leq |L\Delta(\phi_2,\phi_1)\int_{t_0}^{t} ds + M(|\delta_2| - |\delta_1|)|$$

$$\leq |L\Delta(\phi_2,\phi_1)(t - t_0) + M(|\delta_2| - |\delta_1|)|$$

$$\leq |\Delta(\phi_2,\phi_1)La + M(|\delta_2| - |\delta_1|)|$$

$$\leq |\frac{1}{2}\Delta(\phi_2,\phi_1) + M(|\delta_2| - |\delta_1|)|$$

$$\tag{4}$$

Since $a$ is chosen such that $a < min(\frac{c}{M}, \frac{1}{2L})$, hence $La < \frac{1}{2}$. If $\delta_i \sim U(-b,b)$ then $|\delta_i| \sim U(0,b)$. Further $\mathbb{E}_{|\delta_2|-|\delta_1|}(|\delta_2| - |\delta_1|) = 0$ using Lemma 1.3. Thus $\mathbb{E}_{|\delta_2|-|\delta_1|}\Delta(T\phi_1, T\phi_2) \leq \frac{1}{2}\Delta(\phi_1,\phi_2)$ $\qquad\square$

**Lemma 1.3.** $\mathbb{E}_{|\delta_{i+1}|-|\delta_i|}(|\delta_{i+1}| - |\delta_i|) = 0$ since $|\delta_i|, |\delta_{i+1}| \sim$ *Uniform*$(0,b)$ *. The difference of 2 uniform random variables* $U(0,b)$ *follows the standard triangular distribution.*

*Proof.* Let $X_1 = |\delta_{i+1}|, X_2 = |\delta_i|$ are independent $U(0,b)$ random variables. Let $Y = X_1 - X_2$. The joint probability density of $X_1$ and $X_2$ is $f_{X_1,X_2}(x_1,x_2) = 1$ , $0 < x_1 < b, 0 < x_2 < b$ Using the cumulative distribution technique, the c.d.f of Y is

$$F_Y(y) = P(Y \leq y)$$
$$= P(X_1 - X_2 \leq y)$$
$$= \begin{cases} \int_0^{b+y} \int_{x_1-y}^{b} 1dx_2dx_1 & -B < y < 0 \\ 1 - \int_y^b \int_0^{x_1-y} 1dx_2dx_1 & 0 \leq y < b \end{cases}$$
$$= \begin{cases} \frac{b^2}{2} + by + \frac{y^2}{2} & -b < y < 0 \\ 1 - \frac{b^2}{2} + by - \frac{y^2}{2} & 0 \leq y < b \end{cases}$$

$$\tag{5}$$

Differentiating w.r.t y yields the probability distribution function :

$$f_Y(y) = \begin{cases} y + b & -b < y < 0 \\ y - b & 0 \leq y < b \end{cases}$$

$$\tag{6}$$

From the properties of standard triangular distribution, $E_Y[Y] = 0$

$\qquad\square$

**Lemma 1.4.** *The sequence of functions* $\{\phi_k\}$ *obtained using the transformation T as* $\phi_0(t) = \phi_0$, $\phi_{k+1} = T\phi_k$ *converges.*

*Proof.* $\Delta(\phi_2,\phi_1) = \Delta(T\phi_1, T\phi_0) \leq \frac{1}{2}\Delta(\phi_1,\phi_0)$.

Similarly, $\Delta(\phi_3,\phi_2) = \Delta(T\phi_2, T\phi_1) \leq \frac{1}{2}\Delta(\phi_2,\phi_1) \leq \frac{1}{4}\Delta(\phi_1,\phi_0)$.

In general $\Delta(T\phi_{n+1}, T\phi_n) \leq \left(\frac{1}{2}\right)^n \Delta(\phi_1, \phi_0)$

$\implies \Sigma_{n=0}^{\infty} \Delta(T\phi_{n+1}, T\phi_n) \leq \Delta(\phi_1, \phi_0) \Sigma_{n=0}^{\infty} \left(\frac{1}{2}\right)^n$

Since the above sum converges and the completeness of $\Phi$ proves that the sequence $\{\phi_k\}$ converges. $\qquad\square$

**Lemma 1.5.** *T has at most one fixed point with high probability.*

*Proof.* Suppose there were 2 distinct fixed points $\phi_1$ and $\phi_2$. By the definition of a fixed point $T\phi_k = \phi_k$, hence we obtain $\Delta(T\phi_1, T\phi_2) = \Delta(\phi_1, \phi_2)$ which contradicts Lemma 1.2 with high probability as Lemma 1.2 shows that $|T\phi_2(t) - T\phi_1(t)| \leq \frac{1}{2}\Delta(\phi_2, \phi_1) + M(|\delta_2| - |\delta_1|)$. Hence $\Delta(T\phi_1, T\phi_2)$ would not be equal to $\Delta(\phi_1, \phi_2)$ with high probability. $\qquad\square$

## 2 Stiff ODE: Ablation Studies

As discussed in the experiments section of the paper, we use the same setting as the one described in [1].

The ODE is given by

$$\frac{dy}{dt} = -1000y + 3000 - 2000e^{-t}, \tag{7}$$

with the initial condition of $y(0) = 0$. We use the generalized version of the above equation which is $\frac{dy}{dt} = -ry + 3r - 2re^{-t}$. The generalized equation has the same asymptotic behavior as the original. It also reaches a steady state at $y = 3$. Varying $r$ effectively varies the stiffness ratio of the underlying ODE. It thus allows us to analyze the behavior of the various hyperparameters across a wide range of underlying problem difficulty. The experiments were performed using the dormand-prince [2] ODE solver.

Since the experimental setting of the stiff ODE converges in minutes rather than hours, we test out a variety of settings for $b$. We try to identify strategies for choosing the hyperparameter $b$ for the proposed STEER regularization. We also consider the use of a Gaussian distribution in place of the Uniform distribution. We further test out the effect of the capacity on the regularization. We perform experiments to observe the behavior by varying the number of units in the hidden layer of the neural network.

**Effect of varying** $b$**:** As we observe from Fig. 2 as the parameter $b$ varies, we obtain a range of behavior in terms of the MSE error across a wide variety of stiffness ratios. The general trend indicates that larger $b$ such as $b = 0.124, 0.115, 0.085$ have similar behavior and achieve the minimum error in general. Smaller $b$ on the other hand shows behavior similar to standard Neural ODE as is evident from the plots of $b = 0.025, 0.045$. An intermediate value of $b = 0.065$ shows behavior which is better than very small values of $b$ while worse behavior than the large values of $b$. This indicates that higher values of $b$ are better for the proposed STEER regularization as long as $b$ is less than the length of the original interval.

Figure 2: Comparison of the losses for the various choices of b across varying stiffness ratios.

**Effect of varying distributions:** In the proposed STEER regularization we use the Uniform distribution to sample the end point of the integration. The simplicity of the Uniform distribution adds an elegance to the proposed technique. We want to analyze whether it is the inherent stochasticity that makes the technique effective or the particular choice of the Uniform distribution. As we see from

STEER($b = 0.124$)          Standard Neural ODE

Figure 5: Comparison of the losses for the various number of units in the hidden layer for the case of STEER with $b = 0.124$ and a stanard Neural ODE across varying stiffness ratios.

Fig. 3 we observe a similar trend as we had seen in Fig. 2. Greater the stochasticity in the end time the better the performance in terms of MSE.

Delving deeper into the experimental setting of the Gaussian distribution, we sample an end time $t \sim N(t_1, std)$, where $t_1$ was the original end time of the integration and $std$ is the parameter controlling the standard deviation of the Gaussian distribution. Fig. 3 indicates that higher standard deviations $std = 0.124, 0.05$ lead to better performance in terms of MSE. Very small standard deviations start approaching a behavior that is similar to the one shown by standard Neural ODE as exemplified by $std = 0.01, 0.02, 0.03$. It is interesting to note that the transition from $std = 0.03$ to $std = 0.04$ is rather abrupt and shows an intermediate behavior between the smaller and larger values of $std$.

Figure 3: Comparison of the losses for the various choices of standard deviation ($std$) of the Gaussian across varying stiffness ratios.

As we observe from Fig. 3 we see the effective behavior of the best approaches with the Uniform distribution and the Gaussian distribution respectively compared alongside a standard Neural ODE. This plot indicates that the high stochasticity in the cases of the Uniform and Gaussian distributions leads to lower losses in terms of the MSE. We observe a slight advantage of using Uniform distribution rather than the Gaussian distribution.

Although we observe in this case that the Gaussian distribution leads to similar behavior as the Uniform distribution, it comes along with its own implementation challenges. We observe from Fig. 3 that better performance is obtained when $std$ is high. On the flipside when $std$ is high, there might be some sampled values of $t$ which might be less than the initial time $t_0$. To avoid such scenarios, we would have to employ clipping on one side. Clipping on only one side would skew the resulting distribution. Clipping on both sides would add another parameter $clip$. It would decide how much to clip on either side of $t_1$. To simplify the technique and reducing the number of hyperparameters we chose to use the Uniform distribution. While Gaussian distribution with intelligent clipping could be a viable alternative we leave its analysis for future work.

Figure 4: Comparison of a standard Neural ODE along with the Gaussian and Uniform distributions from which the end time $t_1$ can be sampled.

**Effect of varying the network capacity:** To complete our ablation study, we also compare the effect of varying the number of units in the hidden layer of the network. In case of STEER regularization, the reduction of the number of units generally hurts as shown in Fig. 5. The worst performance is

$$\frac{dy}{dt} = -1000y + 3000 - 2000(e^{-t} + e^{-10t}) \qquad \frac{dy}{dt} = -1000y + 3000 - 2000(e^{-t} + e^{-10000t})$$

Figure 6: We experiment with 2 cases which have multiple $e^{-kt}$ in the $\frac{dy}{dt}$ function to be estimated. STEER regularization doesn't totally fail in these scenarios although the performance reduces.

$$\frac{dy}{dt} = -1000y + 3000 - 2000e^{-t} + 1000sin(t) \qquad \frac{dy}{dt} = -1000y + 7000 - 2000e^{-t}$$

Figure 7: Failure cases of the proposed STEER regularization. Adding a periodic term to the equation $\frac{dy}{dt}$ makes it harder to learn. Changing the behavior of the steady state from $y = 3$ to $y = 7$ causes STEER to not reach a viable steady state solution. Note that a standard Neural ODE also fails in these cases

obtained with 250 hidden units. The performance in terms of MSE keeps getting better by increasing the number of hidden units till 500. Increasing further to 700 leads to similar behavior but slight reduction in performance. In case of a standard Neural ODE on the other hand, the behavior is quite erratic as seen in Fig. 5 . Increasing the number of units in the hidden layer doesn't consistently decrease the MSE error.

**Other forms and failure cases:** We analyze other instances of the equation to delve deeper into the effective behavior. When additional terms in form of $e^{-kt}$ are added to the differential equation, our proposed STEER regularization is able to reasonably reach steady state solutions as shown in Fig. 6. We analyze some failure cases in Fig. 7. We observe that when a $sin(t)$ term is added our proposed STEER regularization struggles to fit the periodic behavior. Smaller changes such as changing the steady state from $y = 3$ to $y = 7$, causes our regularization to fail to reach the steady state solution. This indicates that stiff ODEs need further analysis. Alternative techniques or regularizations may be required to deal with harder instances of stiff ODEs.

## 3   Generative Models: Ablation Studies

We analyze the effect of varying $b$ in case of continuous normalizing flow based generative models. We conduct ablation studies on the setting of RNODE [3] since it converges much faster. Multiple possible values of $b$ could be experimented, since convergence is fast. As we observe from Table 1

|  | MNIST | |
|---|---|---|
|  | BITS/DIM | TIME |
| $t_1 : t_1 + b = t_1^{original} = 1$ | | |
| FFJORD RNODE + STEER ($b = 0.5$) | 0.971 | 16.32 |
| FFJORD RNODE + STEER ($b = 0.375$) | 0.973 | 17.13 |
| FFJORD RNODE + STEER ($b = 0.25$) | 0.972 | 19.71 |
| FFJORD RNODE + STEER ($b = 0.125$) | 0.973 | 22.32 |
| $t_1 : t_1 = t_1^{original} = 1$ | | |
| FFJORD RNODE + STEER ($b = 0.5$) | 0.974 | 25.81 |
| FFJORD RNODE + STEER ($b = 0.375$) | 0.98 | 25.72 |
| FFJORD RNODE + STEER ($b = 0.25$) | 0.971 | 25.23 |
| FFJORD RNODE + STEER ($b = 0.125$) | 0.976 | 24.32 |

Table 1: Comparison of $b$ in case of Continuous Normalizing Flows. Greater values of $b$ lead to faster convergence. Altering $t_1$ such that $t_1 + b = t_1^{original}$ leads to faster convergence times.

there is a general trend which is similar to the one we observed in the case of stiff ODEs. The greater the stochasticity due to a larger value of $b$, the faster the convergence time. The best results were obtained by using $b = 0.5$. In case of generative modeling, to obtain faster convergence, the end time $t_1$ had to be constrained such that $t_1 + b = t_1^{original}$. The original ending time was $t_1^{original} = 1$ for the experiments in Table 1. The bottom half of Table 1 demonstrates that if the $t_1$ is not altered i.e. $t_1 = t_1^{original}$, faster convergence is not observed.