[Reviews · NeurIPS 2020]

Review 1

Summary and Contributions: In this paper, the authors propose a simple regularization for neural ODE. The idea is to add a uniformly distributed random noise to the final time of the initial value problem during training.

Strengths: The proposed model has the merit of simplicity. It's easy to understand and implement. The experimental study is well designed and extensive.

Weaknesses: I fail to fully understand why the proposed regularization helps. In many applications of neural ODE, for example in generative models, the final time t_1 is somewhat arbitrary. The proposed regularization is effectively changing the final time from t_1 to (t_1 - b). This is also supported by Figures 1, 2, and 3. I understand that it is unfair to ask for a comparison with Finlay et al. (2020) since it is recently published. But I think their method is much more principled and theoretically founded.

Correctness: The claims and method are correct. The empirical methodology is consistent with how existing models are evaluated.

Clarity: The paper is well written and easy to understand.

Relation to Prior Work: As mentioned above, it would be great if the authors can further elaborate how their method is different from Finlay et al. (2020).

Reproducibility: Yes

Additional Feedback:


Review 2

Summary and Contributions: In this paper, the authors propose a novel regularization technique for Neural ODEs, where the end time of solving ODEs during training is sampled randomly. They show the validity of this approach based on the existence of unique solutions. And, they empirically validate the proposed technique to work expectedly on normalizing flows, time series models and image recognition tasks.

Strengths: The paper propose a (minor but) novel technique that is potentially useful, where the existence of unique solutions is assured theoretically. In the experiments, their proposed technique is shown to give some improvements over test errors in several experimental setups under comparison with the representative recent models.

Weaknesses: First of all, I think the efficacy of the proposed technique in practice is limited. As the authors claim, the end time of solving ODEs during training would have effects to computational costs and test accuracy somehow. I agree with this point intuitively (although I have some doubt on the point that the computational cost is reduced much from this perspective). However, it is difficult to judge whether their technique, where the end time of solving ODEs during training is sampled randomly, improve in principle these by using this effect from the current descriptions of this paper. I would appreciate if the authors address how this regularization affects positively on test accuracy (through the improvement over stiffness?).

Correctness: Although I do not go into the details, the derivations of the equations and the related logics in the paper look correctly described.

Clarity: The clarity of the paper is OK. It contains necessary information to understand the contents.

Relation to Prior Work: The paper describes clearly the relation with existing works and their contributions upon those.

Reproducibility: Yes

Additional Feedback:


Review 3

Summary and Contributions: The paper presents a regularization technique for training Neural ODE models. The paper presents a theorem, tests the methodology on standard problems, and discusses some connection to stiff problems.

Strengths: The topic is of very high relevance to the community as there is currently a spike in interest in Neural ODE models. The empirical results are promising.

Weaknesses: I have trouble following the rationale of the theory, both for the theorem as well as the discussion of stiffness.

Correctness: - I don't understand the necessity of the theorem: either the solution to the Neural ODE is well-defined over [t_0, t_1 + b] and there is nothing to proof as we fall back to the deterministic case, or it isn't , but in this case there shouldn't exist a meaningful gradient in the first place. - I also don't understand the proof of the theorem: why is delta_k drawn in each iteration of the Piccard? in the training, T is only drawn after a theta-parameter update has been conducted. So within each theta iteration, the integration interval is fixed, no? - I have reservations regarding the stiffness discussion. This might be slightly off, as I haven't been able to fully follow the synthetic study design (see below). That being said: the problem of stiffness has a rich history in the study of numerical ODE solvers. Sect. 3.4 is a bold simplification of the subject manner without any external reference to the numerical analysis literature. Furthermore, there is no rationale presented as to why the presented method should be able to better deal with stiffness. I'd even argue that the paper's intuition could be used to argue why STEER shouldn't be able to handle stiffness: STEER encourages to find solutions that achieve the final state earlier which necessitates faster transients, while occasionally asking for solutions t_1+b asks for slow transients. Thus, I'd expect stiff behavior of the found solutions in the interval [t_1 - b, t_1 + b]. - Furthermore: if I understood the stiffness experiment halfway correctly, the Neural ODE is used to fit a particular trajectory. However, in all Neural ODE applications that I am aware of until now, the model is always ill-posed in the sense that there is never an unique vector field solution to the problem, but rather many viable dynamics that could fit the data. Thus, I don't think that the experiment is a clear indication what is happening with respect to stiffness in Neural ODE training.

Clarity: - I haven't been able to understand the problem setup of Sect. 4.4, neither from the main text nor from the appendix. I believe that the exact training and evaluation method are presented in the paragraph starting in line 267, but I have not understood the description. As a reference, a 990 page text book is cited without a hint to any specific subsection.

Relation to Prior Work: The discussion of related work with respect to Neural ODEs is adequate. Regarding the discussion of stiffness, the authors should consider refering to the discussion in https://arxiv.org/abs/0910.3780 https://rd.springer.com/article/10.1007/s10543-014-0503-3 https://books.google.de/books/about/Solving_Ordinary_Differential_Equations.html?id=m7c8nNLPwaIC&redir_esc=y or similar work (but also see comment below).

Reproducibility: No

Additional Feedback: While I have raised severe objections, I still believe that the method itself may have strong merits. Please consider the following questions and suggestions: a) If the authors deem the theorem really necessary, it needs to be made clearer why the regular Picard-Lindelöf theorem does not apply. Maybe I misunderstood something on a very fundamental level. If not, simply consider removing the section. b) The discussion, rationale and experiments regarding stiffness does not present a coherent picture to me and I have severe objections for acceptance If the authors chose to keep this content, I would require significant changes and very convincing explanations during the rebuttal. However, I don't think the method necessarily requires a stiffness discussion. c) If either or both sections are removed, may I suggest to add more experimental details that would certainly be of interest to the readers: how are training times affected for time series and feedforward models? Is there a dependence between suitable parameter ranges b and different solvers/model architectures? Have you also tried to apply the stochastic term during evaluation? Maybe the authors could even try to include fixed step solvers (ResNets with stochastic depth)? From my background in numerical analysis, it currently seems as if the author felt the need to include more theory to warrant publication, but these parts raise more concerns than bring valuable insights. On the other hand: a more nuanced empirical view on the method would probably find a big audience and I would gladly support such a submission, even if theoretical questions are left for future work. Post-rebuttal update: I thank the authors for their feedback and I'm happy to hear that I could provide good hints. Big changes have been promised (the removal of Sect. 3.3 and at least some more nuanced description of Sects. 3.4+4.4 with the remaining space being filled by more empirical evaluations). I am willing to give the authors the benefit of the doubt and raise no more objections, but I will also not actively seek acceptance at this time. Thus, I raised my overall score, but lowered my confidence.


Review 4

Summary and Contributions: The paper proposes a regularization technique for Neural ODE models, which speeds up the notoriously slow training of these models, while simultaneously improving their performance on a range of tasks. The technique is simple and complementary to existing approaches.

Strengths: The work proposes a simple and effective regularization technique that addresses an important problem in practical applications of Neural ODE models - their computational complexity. While not providing an ultimate solution - even with the proposed technique these models are still considerably slower to train than fixed-depth neural networks - this work makes a step towards solving this issue, and would be interesting to a wide audience. Effectiveness of the technique is showcased on a number of tasks (density modeling, image classification and time series modeling) with convincing results. Analysis of toy Gaussian densities and the stiff ODE example further strengthen the results and give insight into the effect of this regularization technique. Finally, the paper also provides theoretical grounding for the existence of a solution to a neural ODE with their regularization technique applied.

Weaknesses: It is not entirely clear from the paper how difficult choosing the right value of the regularization term b is in practice. It would be helpful if the authors could share results of such experiments for the density modeling or classification experiments. Such results are important to gauge the practical value of the proposed regularization technique.

Correctness: Yes.

Clarity: The paper is well-written and well-motivated. One thing I missed was the comparison of the Neural ODE results to results of fixed-depth methods. Including these would make it easier for the readers to understand the value of using Neural ODEs for certain tasks compared to other types of models. Also a few minor comments/questions below: * In equation 5 are eigenvectors meant to be complex or real? * Please mention dataset details in Tables 1 and 5 (i.e. CIFAR10, ImageNet32). * What are the units of time in Table 1? * Please check that the following sentence is correct: “Indeed these datasets break the basic assumptions of a standard RNN, namely that the data are not necessarily collected at regular time intervals” * In Table 4 add space before “STEER” in rows 5 and 6

Relation to Prior Work: Yes.

Reproducibility: Yes

Additional Feedback: ======= POST-REBUTTAL UPDATE ======= I would like to thank the authors for their response, for providing additional details and committing to making some changes to the manuscript. I believe the manuscript will improve as a result.

[Author Response · NeurIPS 2020]

We thank the reviewers and are happy that they found the solution simple (R1,R4) and the empirical results promising (R1,R2,R3,R4) in a field of high relevance to the community(R3).

The questions raised by R3 are valid and we thank R3 for raising them since it would aid in making the text better. We thank R3 for pointing us to the references of stiff ODEs in the field of numerical analysis. We would definitely add these as references. The experimental setting for section 4.4 was similar to the one used in the examples/ode_demo.py file of the official github repository of neural ode. Instead of using a 2-D ODE we used a 1-D ODE to be able to better visualize and analyze the behavior. We used Eq 12 in the paper as mentioned in the numerical methods book by Chapra et al on page 752. It was surprising that a Neural ODE was able to model a 2 dimensional model with ease while the stiff equation was extremely hard for the Neural ODE to model. We agree that we have barely scratched the surface in terms of stiff ODEs. We had mentioned in lines 287-291 explaining that it was an empirical observation regarding the failure of neural ODE based models where STEER was able to somehow work better. The text wanted to highlight the problems of stiff ODEs and demonstrate that it could become a real challenge moving forwards as the area of Neural ODE matures. We would be open to removing the section until we understand more about the reason why STEER performs better in those circumstances. As R3 points out the method still would have merits without the stiff ODE section.

The concerns raised regarding the proof is also valid on closer inspection. The proof, though valid does not add much in the discussion of the proposed STEER method since T is only drawn after a theta-parameter update has been conducted. Our rationale was based on the fact that Picard's iteration enables us to prove existence of a unique solution for Neural ODEs and the modified Picard's iteration enables us to show the existence of a unique solution for one with changes in the end time. However, modifications to the Picard's iterates on closer inspection are not the exact same modifications introduced by STEER due to the random sampling involved. We would be open to removing the discussion of the proof from the paper.

As R4 pointed out, the eigenvectors can be complex or real. The ratio of the magnitudes of the real parts of the eigenvalues defines the stiffness ratio. We will mention dataset details as pointed out for Tables 1 and 5. The units of time of table 1 is hours. The sentence can be better phrased as RNNs have also been used for irregularly sampled timeseries models with good results.

R4 and R3 have pointed out the comparison to fixed depth solvers such as resnets. Extensive comparisons between resnets and ODEs with similar number of parameters were made in Neural ODE Chen et al. (2018) and Augmented Neural ODE Dupont et al. (2019). STEER doesn't provide a huge performance improvement in terms of accuracy over these 2 techniques. We would however include the results for a fair comparison. With respect to the optimum values of $b$ as R4 and R3 have pointed out. We have shown some experiments in the supplementary regarding the values of b for optimal performance. The general observation has been that the greater the value of b the better the performance. In case of feedforward models with fixed limits of integration $[0, 1]$ the best performance was obtained when $b = 0.99$. In the case of generative models with continuous normalizing flows, however there were some numerical issues when we tried values of b as high as feedforward models. The best results while also ensuring numerical stability was obtained with $b = 0.5$ with fixed limits of integration $[0, 1]$. In case of irregularly sampled timeseries models, since the points are irregularly sampled, hence the limits of integration change every time. In such a case rather than explicitly tuning $b$ we tune the $\epsilon$ parameter as mentioned in line 207 the main text of the paper. Similar to the observations in the preceding cases, the best results are obtained using $b$ as high as possible. In this case it implies $\epsilon$ as low as possible without numerical instability. The best results were obtained with $\epsilon = 0.05$.

As R1 has rightly pointed out, the integration times are arbitrary in the case of Neural ODEs. The dynamics of these models rearrange to similar qualitative behaviors even when the integration times are varied by instance. We observe that STEER changes the behavior of these models in qualitatively meaningful ways as shown in Figure 1. The principled work by Finlay et al. (2020) is based on concepts of optimal transport. We have shown via experiments on generative modeling (Table 1) that our technique can be used to augment their technique to obtain even faster convergence times.

Neural ODE          STEER

Figure 1: A simple addition of $t$ to $sin(t)$ changes the qualitative performance of Neural ODE vs STEER.

R2 points out the rationale behind the increase in the performance in case of timeseries models and feedforward models. Previous studies Dupont et al. (2019) have corroborated that simpler dynamics leads to better solutions. The temporal regularization is intended to improve the dynamics for neural ODE flows as demonstrated via multiple experiments. It could include unintended improvements in accuracy that we observe in Tables 2,4 and 5 and Figure 1. We intend to explore these aspects in future work.

[Meta-Review · NeurIPS 2020]

This submission proposes a new regularization scheme for ODE NNs. The method is simple and new and alleviates a key weakness of these models. The theoretical justification and experimental validation could be stronger.